# Towards Physical, Imperceptible Adversarial Attacks via Adversarial Programs

## Abstract

Adversarial examples were originally defined as imperceptible perturbations which cause a deep neural network to misclassify. However, the majority of imperceptible perturbation attacks require to perturb a large number of pixels across the image and are thus hard to execute in the physical world. Existing physical attacks rely on physical objects, such as patches/stickers or 3D-printed objects. Producing adversarial patches is arguably easier than 3D-printing but these attacks incur highly visible perturbations. This raises the question: *is it possible to generate adversarial examples with imperceptible patches?* In this work, we consider adversarial multi-patch attacks, where the goal is to compute a targeted attack consisting of up to $K$ patches with minimal $L_2$ distortion. Each patch is associated with dimensions, position, and perturbation parameters. We leverage ideas from program synthesis and numerical optimization to search in this large, discrete space and obtain attacks that are competitive with the C&W attack but have at least 3x and up to 10x fewer perturbed pixels. We evaluate our approach on MNIST, Fashion-MNIST, CIFAR-10, and ImageNet and obtain success rate of at least 92% and up to 100% with at most ten patches. For MNIST, Fashion-MNIST, and CIFAR-10, the average $L_2$ distortion is greater than the average $L_2$ distortion of the C&W attack by up to 1.2.

## 1 Introduction

Adversarial examples were originally defined as imperceptible perturbations added to correctly classified inputs which cause a deep neural network (DNN) to misclassify (Szegedy et al., 2014; Goodfellow et al., 2015). Since then, many works have shown how to generate imperceptible adversarial examples (e.g., Carlini & Wagner (2017); Moosavi-Dezfooli et al. (2016); Madry et al. (2018)). A main motivation of this line of works is to expose the level of vulnerability of DNN-based systems. However, these attacks often cannot be executed in the physical world, for example against DNNs used by autonomous vehicles (Lu et al., 2017). This gave rise to works that focus on physical attacks, executed by adding patches or stickers to objects or cameras (Brown et al., 2017; Eykholt et al., 2018; Li et al., 2019) or 3D-printed objects (Athalye et al., 2018; Sharif et al., 2018). While 3D-printing enables the attacker to generate imperceptible adversarial examples, it is not the case for patch/sticker attacks. These attacks incur highly visible perturbations. However, in some settings (e.g., traffic sign attacks), patch attacks are easier to execute than 3D-printing attacks. This raises the question: *is it possible to generate adversarial examples with imperceptible patches?*

We consider a new space of targeted attacks expressed as patch sequences. The patches we consider are of rectangular shape and uniformly perturb their associated pixels according to scale and shift factors. Given an input, a DNN, and a target label, the attacker's goal is to compute a patch sequence resulting in a targeted adversarial example of minimum distance to the original input. That is, the attacker chooses the patches' position, dimensions, perturbation factors, and the number of patches such that the output of the patch sequence is an imperceptible adversarial example. Because the goal is to generate a sequence of patches, each is constrained by a structure, it is natural to view each patch as a *program instruction* and the overall problem as an instance of Syntax-guided Synthesis (SyGuS) (Alur et al., 2013), where a high-level requirement (e.g., computing an adversarial example) has to be implemented in a given programming language (e.g., the patch sequence). A common challenge of SyGuS instances is looking for a solution (i.e., a program) in a discrete space. Unlike common SyGuS instances, our instructions (i.e., patches) are defined also over continuous variables

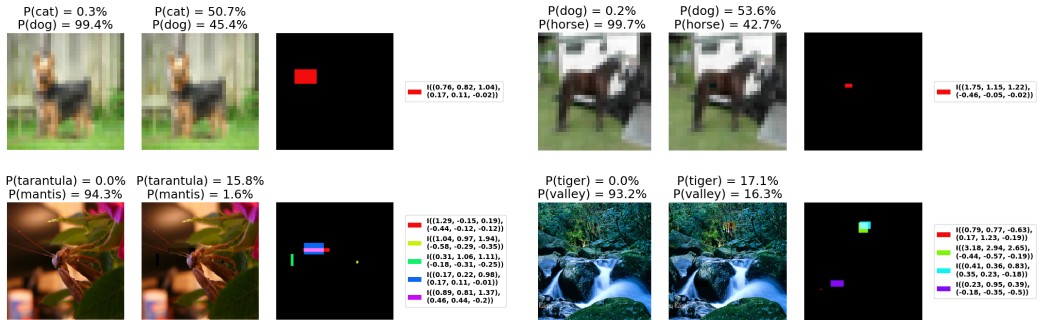

Figure 1: Examples of APSyn's attacks for CIFAR-10 (top) and ImageNet (bottom). Each triple consists of an image (left), the adversarial program (right), and the adversarial example returned by the program (middle). Programs are depicted as images showing the patches' position and dimensions; the legend shows the scale and shift factors of each patch, one per color channel.

(the perturbation factors). Thus, our setting is not easily amenable to standard SyGuS approaches, neither to standard numerical optimization approaches (because of the discrete variables).

We propose APSyn, an **a**dversarial **p**rogram **syn**thesizer, to compute patch sequences. APSyn builds on enumerative program synthesis which enumerates programs from the shortest to the longest until obtaining a solution (Udupa et al., 2013). This approach guarantees to return the shortest solution among all solutions, which is desirable when aiming at simplifying the physical execution of the attack. While enumerative synthesis is useful, it is not sufficient by its own, as there are infinitely many ways to instantiate a patch (up to computer representation). To search for the optimal patch parameters, we leverage the loss proposed by Carlini & Wagner (2017). To optimize this loss for patch sequences, we propose a gradient estimate for the discrete variables using zeroth order optimization. We further allow discrete variables to take real values to avoid skipping optimal solutions.

Our evaluation results for MNIST, Fashion-MNIST and CIFAR-10 show that the success rate of APSyn is at least 96%, when allowed up to five patches. The $L_2$ distortion of these attacks is on average greater by up to 1.2 than the C&W's $L_2$ distortion, while the $L_0$ distortion is smaller by at least 3x and up to 10x compared to the C&W's attack. We further show that APSyn produces attacks which generalize better: at least 2.7x and up to 23x more than the C&W attack. We believe the higher generalization rate stems from the structure imposed on our attacks. For ImageNet, APSyn's success rate is 92% when allowed up to ten patches. Figure 1 shows several attacks that APSyn synthesized along with visualizations of the patch sequences.

To conclude, our main contributions are:

- A new class of adversarial attacks consisting of multi-patch sequences.
- An effective algorithm to compute multi-patch attacks combining enumerative synthesis and numerical optimization. The latter relies on gradient estimation of the discrete factors.
- An extensive evaluation of our approach on four datasets. Results show that our attack is competitive with the C&W attack, in terms of the success rate and $L_2$ distortion, but has significantly lower $L_0$ distortion and is thus easier to physically execute.

## 2 ADVERSARIAL PROGRAMS

In this section, we provide the terms and notation used throughout the paper and define our problem.

We are given a classifier $N : [0, 1]^{d_1 \times d_2} \rightarrow [0, 1]^C$, mapping a two-dimensional image to a probability vector over $C$ classes. We note that all our definitions extend to colored images by considering each color channel separately. We assume a white-box access to the classifier. We denote by $class(N(x))$ the classification of input $x$ by $N$, i.e., the index with the maximal probability. We are also given an input $x$ with class $c$, which $N$ classifies correctly, i.e., $class(N(x)) = c$, and a target adversarial label $t \neq c$. The goal of the attack is to compute an input $x'$ which is similar to $x$ but is classified as $t$. Commonly, this is phrased as constrained optimization, where the goal is to compute

a perturbation matrix $M \in [0,1]^{d_1 \times d_2}$ such that the adversarial example $x' = x + M$ is classified as $t$ ($class(N(x')) = t$) and the $p$-norm of the perturbation matrix $||M||_p$ is minimal.

In this work, we aim at adding structure to adversarial attacks with the goal of obtaining attacks which are easier to physically execute, are more explainable, and generalize better. We draw inspiration from program synthesis where the goal is automatically generate a program over a given programming language satisfying a set of requirements (Gulwani et al., 2017). In particular, we view our problem as an instance of Syntax-guided Synthesis (SyGuS) (Alur et al., 2013), where the goal is to implement a high-level specification in a given programming language, typically constrained and small. Having a program has various advantages: it can be used on other inputs, the implementation can provide an alternative characterization of the user intent, or even provide a more efficient implementation than the user initially had. These applications are useful within our scope: our adversarial programs can be executed on other inputs in a negligible time and our programs expose the (almost imperceptible) shapes that cause the DNN to change its decision.

**Adversarial programs**    We introduce a programming language for adversarial multi-patch attacks. Our language consists of sequences of patch instructions from a family of patches $\mathcal{I}$. Our patch family generalizes the perturbation matrices discussed before to scaling and shift perturbations over rectangular masks. A rectangular mask is a tuple consisting of a starting and ending position $p = (r^s, c^s, r^e, c^e)$ capturing a single rectangle in the input. A patch instruction (or simply, a patch) $I_{\alpha,\beta,p} \in \mathcal{I}$ is associated with a mask $p$, scale factor $\alpha$, and a shift factor $\beta$, and takes the form:

$$I_{\alpha,\beta,p}(y) = y \rightarrow \text{lambda } v : (\alpha \cdot v + \beta) \text{ if } idx(v) \in p$$

where $\alpha, \beta \in \mathbb{R}$ and $p \in (\{1,\dots,d_1\} \times \{1,\dots,d_2\})^2$. Given $y \in [0,1]^{d_1 \times d_2}$, $I_{\alpha,\beta,p}(y)$ maps every value $v$ in $y$ to $\alpha \cdot v + \beta$, if its index is inside the mask. Values outside the mask remain as are. For value $v$ with $idx(v) = (r_v, c_v)$ and mask $p = (r^s, c^s, r^e, c^e)$, we say $v$'s index is inside $p$ if $r^s \le r_v < r^e$ and $c^s \le c_v < c^e$.

**Problem definition**    Given a network $N$, an input-output pair $(x, c)$, a target class $t$, and a program length $k$, our goal is to compute a program $P$ over $\mathcal{I}$ with $k$ or fewer patches such that $N$ classifies $P(x)$ as $t$ and $P(x)$ is as close as possible to $x$ with respect to the $L_2$ norm:

$$\min ||P(x) - x||_2$$
$$s.t.$$
$$P(x) = I_{k'}(\dots(I_1(x))) \quad I_1,\dots,I_{k'} \in \mathcal{I}, k' \le k$$
$$class(N(P(x))) = t$$

Note that we allow patches to overlap, which may lead to perturbation compositions.

**Designing a solution**    If we set aside the constraint that $P$ is a program over $\mathcal{I}$, a common approach to solve the above constrained optimization is by relaxing the attack constraint. For example, Carlini & Wagner (2017) propose to relax it to a term which is then minimized along with the norm:

$$\mathcal{L}(P) = \max(\max_{i \neq t} N(P(x))_i - N(P(x))_t, 0) + \lambda \cdot ||P(x) - x||_2$$

where $\lambda$ is a hyper-parameter balancing the two optimization goals. If $P(x)$ were differentiable, gradient descent could minimize this loss. If the size of $\mathcal{I}$ were relatively small, enumerative search would have been feasible. However, neither of these hold. We thus approach this search by combining the two ideas. First, we introduce a gradient estimation approach for the discrete variables, and then let Adam (Kingma & Ba, 2015) use it to minimize $\mathcal{L}(P)$. Second, to compute a shortest patch sequence and search more efficiently, we employ enumerative synthesis and gradually synthesize the adversarial program's patches. We explain these steps in the next two sections.

## 3    GRADIENT ESTIMATION OF THE PATCHES' DISCRETE PARAMETERS

In this section, we explain our approach to search for an optimal patch attack consisting of exactly $k$ patches from $\mathcal{I}$. Unlike previous patch attacks, our approach jointly optimizes over all the patches' parameters – position, dimensions, and perturbation factors. Since we allow patches to overlap, this optimization may lead to complex patterns, despite the rectangular shape of the patches. Our

approach builds on the C&W attack, which minimizes the loss defined in the previous section. Since our attacks involve discrete variables, to numerically optimize this loss, we define "gradients" for these variables. The idea is to estimate the partial derivatives of the discrete variables using zeroth order optimization. We note that we do not claim that the partial derivatives exist, but rather that estimating them can guide the optimizer how to change a discrete variable so that our loss decreases.

**Gradient estimation**    Given a program $P$, an input $x$, and the target label $t$, the partial derivatives of the real-valued variables (the scale and shift parameters $\alpha$, $\beta$) are computed as usual. The partial derivative of a discrete variable $w$ is estimated by adapting the standard definition of the derivative. As a first step, we estimate the derivative in each direction:

$$\mathcal{L}(P)_w^+ \triangleq \frac{\partial \mathcal{L}(P)^+}{\partial w} = lim_{h \to 0^+} \frac{\mathcal{L}(P[w \to w + h]) - \mathcal{L}(P)}{h} \cong \mathcal{L}(P[w \to w + 1]) - \mathcal{L}(P).$$

The last step is obtained by replacing $h$ with the closest discrete value to 0. The notation $P[w \to w + h]$ denotes the program identical to $P$ except that $w$ is increased by $h$. Similarly, $\mathcal{L}(P)_w^- \cong \mathcal{L}(P) - \mathcal{L}(P[w \to w - 1])$. These "derivatives" simulate the role of the gradient in optimization. If $\mathcal{L}(P)_w^+$ and $\mathcal{L}(P)_w^-$ are negative, then increasing $w$ will decrease the loss and thus we define the gradient as $\mathcal{L}(P)_w^+$. If $\mathcal{L}(P)_w^+$ and $\mathcal{L}(P)_w^-$ are positive, then decreasing $w$ will decrease the loss, and thus we define the gradient as $\mathcal{L}(P)_w^-$. If $\mathcal{L}(P)_w^+$ is negative and $\mathcal{L}(P)_w^-$ is positive, we define the gradient based on their magnitude; if $\mathcal{L}(P)_w^+$ is positive and $\mathcal{L}(P)_w^-$ is negative, the gradient is zero.

**Optimization**    Using this estimation, we can minimize our loss using Adam (Kingma & Ba, 2015) and Hill Climbing. Each optimization step begins with an optimization of the discrete variables followed by a standard, joint, Adam step for the continuous variables. For the discrete variables, we perform Hill Climbing and Adam. That is, for each discrete variable, we separately estimate its gradient and update the variable according to Adam's update steps. If the discrete variable is assigned an out-of-bounds value, its previous value is restored. We note that the optimizer has two optimization steps: $\eta_c$ for the continuous variables and $\eta_d$ for the discrete variables. In practice, it is best that $\eta_d \approx 100 \cdot \eta_c$. To reduce execution times, the optimization terminates if within $T$ iterations, the loss has decreased by less than $p\%$, where $T$ and $p$ are hyper-parameters. At the end, our adapted Adam optimizer returns a program with the optimized continuous and discrete parameters.

**Real-value relaxation**    Although patches' position and dimensions are discrete variables, we observe that it is better to allow them to take real values during optimization. This relaxation enables the optimizer to use small step sizes, and thereby reach solutions which may have been missed if the variables would only take discrete values. With this relaxation, the discrete variables are rounded whenever $P$ is invoked on an input (as part of the optimization or when executing the attack).

## 4    APSYN: ADVERSARIAL PROGRAM SYNTHESIZER

In this section, we present APSyn, our system for synthesizing adversarial programs. APSyn aims at obtaining a minimal-sized patch sequence in order to compute the simplest attack. To this end, we employ a simple yet effective program synthesis approach called enumerative program synthesis, shown successful for many syntax-guided synthesis instances. An enumerative synthesizer considers programs of increasing size and checks whether they satisfy the specification. At each step, the synthesizer considers all programs of size $k'$. In our context, it is not feasible to consider every patch sequence of size $k'$. Instead, we search for useful programs of size $k'$ using the optimization described in the previous section. Since the optimization can return a local optimum, we invoke the optimizer multiple times with different initialization, and continue with the best solution. Algorithm 1 shows the operation of APSyn. APSyn synthesizes the adversarial program patch-by-patch. At iteration $i$, it constructs the program $P_i$, by considering $m$ patch candidates to extend the previous program $P_{i-1}$. For each candidate patch instruction $I_j$, APSyn jointly optimizes over the variables in the extended program $P_{i-1}::I_j$, as described in Section 3. The program $P_i$ is defined as the optimized program that has the best loss out of the $m$ candidates and $P_{i-1}$[1]. We note that it is possible to synthesize a full program from the beginning (instead of the enumerative search), but it requires many more iterations to converge to a successful adversarial program (see Section 5).

---

[1]This is a simplification, APSyn prefers a program that generates an adversarial example over one with lower loss which does not generate an adversarial example.

---

**Algorithm 1:** APSyn $(N, x, t)$

---

**Input:** A classifier $N$, an input $x$, a target label $t$.
**Output:** An adversarial program over $\mathcal{I}$.
$P_{best} = P_0 = [] \; ; L_{best} = \infty$
**for** $i = 1; i \leq k; i + + $ **do** // Synthesize programs of increasing sizes
  **for** $j = 1; j \leq m; j + +$ **do** // Consider $m$ candidates
    $I_j = \mathrm{init}()$
    $P = \mathrm{Adam}(P_{i-1} :: I_j, N, x, t)$
    **if** $\mathcal{L}(P) < L_{best}$ **then**
      $\lfloor$ $L_{best} = \mathcal{L}(P); P_{best} = P$

  $P_i = P_{best}$
**return** $P_{best}$

---

**Initialization**   A patch is initialized by sampling the dimensions (length and width) from a uniform distribution and sampling values for $\alpha$ and $\beta$ from Gaussian distributions. To initialize the starting position, we build on feature ablation. That is, APSyn considers every possible feasible starting position (i.e., one that together with the given dimensions fit within the image boundaries). For each possibility $(r^s, c^s)$, it computes the potential decrease of the loss, if we use that position in the new patch without any optimization. Overall, it initializes the starting position to the position minimizing $\mathcal{L}(P_{i-1} :: I_{r^s,c^s,r^e,c^e,\alpha,\beta}(x)) - \mathcal{L}(P_{i-1}(x))$, where the ending position is inferred from the starting position and the patch's dimensions. To reduce the execution overhead of checking every possible starting position, we examine starting positions based on a stride hyper-parameter.

## 5   EVALUATION

In this section, we evaluate APSyn. We implemented APSyn in Python using PyTorch. Our implementation supports GPU parallelization. It will be made available, for reproducibility, along with the experiments. Experiments ran on an Ubuntu 18.04.5 OS on a dual AMD EPYC 7742 server with 1TB RAM and eight NVIDIA GeForce RTX 2080 Ti GPUs. Experiments ran for MNIST, Fashion-MNIST, CIFAR-10, and ImageNet. Experiments were parallelized: between 4-7 APSyn's invocations on each GPU, depending on the dataset. For MNIST and Fashion-MNIST, we trained MnistNet models (186K parameters). For CIFAR-10, we trained a ResNet-18 model (11.5M parameters). For ImageNet, we imported a ResNeXt-50-32x4d model (25M parameters). We next provide the hyper-parameters. The scale and shift parameters were initialized by sampling from Gaussian distributions with mean 1 and 0 (respectively) and variance 0.3. Patch dimensions were sampled from a uniform distribution with range [5,15] (for ImageNet, [15,30]), and the starting position stride was 3 (for ImageNet, 10). The balancing factor $\lambda$ was 1 (for ImageNet, 0.05), the step sizes were $\eta_c = 0.01$ (for ImageNet, 0.05) and $\eta_d = 1$ (for ImageNet, 2.5). The maximal number of iterations per optimization was 300 (for ImageNet, 400) and the stopping condition checked whether during the last 25 iterations (for ImageNet, 200) the loss has decreased by less than $0.5\%$.

**Benchmarks and metrics**   For each dataset, our benchmark consists of three images for each class (for ImageNet, we focus on ten classes). In each experiment, we run APSyn for every image and every possible target class (for ImageNet, we focus on three target classes). In total, our benchmarks consist of 270 attack specifications for each dataset (90 for ImageNet). We measure the success rate (fraction of programs that returned adversarial examples), $L_2$ distortion ($L_2$ distance between the adversarial example and the input), and $L_0$ distortion ($L_0$ distance). We report execution time in minutes. We run every APSyn attack on all inputs in the test set whose label is $c$ (the input's true label). We measure generalization rate (fraction of inputs for which the program returned adversarial examples) and generalization $L_2$ distortion (average $L_2$ distortion on the test set). Execution times and all distortion metrics are reported only for successful attacks. As a baseline, we compare to the C&W attack (on which we build our loss function), which is run in batches of 32 attacks.

**Candidates**   We begin with studying the effect of the number of candidates $(m)$ on the success rate, $L_2$ distortion and generalization rate. In this experiment, we focus on three datasets, and set the

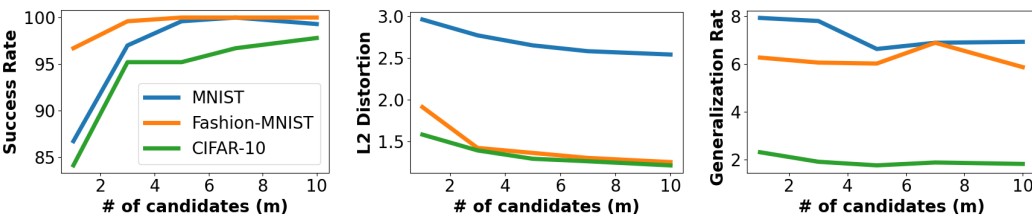

Figure 2: Effect of number of candidates on the success rate, $L_2$ distortion, and generalization rate.

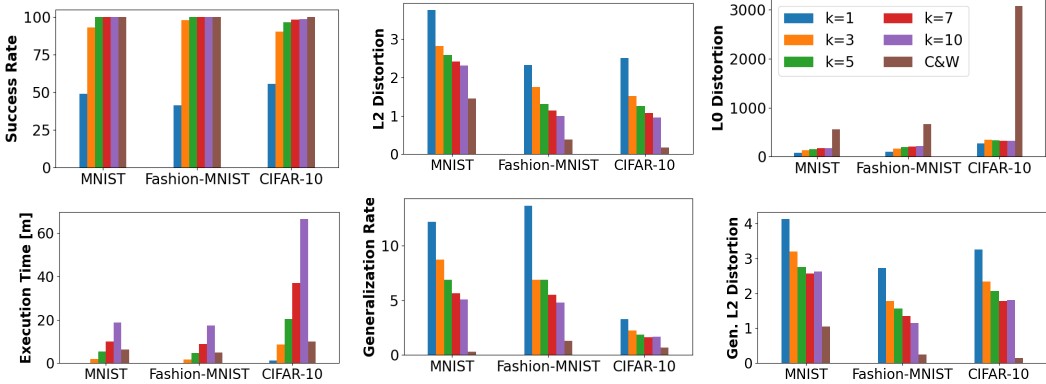

Figure 3: Effect of number of patch instructions on APSyn's success rate, $L_2$ distortion, $L_0$ distortion, execution time, generalization rate, and generalization $L_2$ distortion on different datasets.

maximal number of instructions to $k = 5$. Figure 2 shows the results as a function of the number of candidates $m \in \{1, 3, 5, 7, 10\}$. As expected, as the number of candidates increases, the success rate increase and the $L_2$ distortion decreases. However, the gain in adding more candidates is reduced as the number of candidates increases. From here on, we continue with $m = 7$ candidates.

**Program length** We next study the effect of the maximal number of patches ($k$) on the success rate, $L_2$ distortion and the execution time of APSyn. Figure 3 shows the results as a function of the maximal number of patches $k \in \{1, 3, 5, 7, 10\}$. As a baseline, the figure also shows the results of the C&W attack. As expected, as the number of patch instructions increases, the success rate increases, but with three patches APSyn usually obtains an almost perfect success rate (similarly to C&W). The reason to include more than three patches is mainly to reduce the $L_2$ distortion, as shown by the figure. Naturally, more patches mean longer execution times. We observe that the best balance is obtained for five patches. A unique feature of multi-patch attacks is that in addition to minimizing the $L_2$ distortion, they also attempt to reduce the $L_0$ distortion, both by favoring fewer patches in the attack and also by allowing overlapping patches. The figure shows how significant is the difference between the $L_0$ distortion of APSyn and C&W. The lower $L_0$ distortion may also indicate about the simplicity of physically executing our attacks compared to attacks with high $L_0$ distortion. In terms of generalization, we see that the shorter the program the higher the generalization rate. This may be attributed to Occam's Razor: the simpler the attack the better its generalization rate. This also aligns with the very low generalization rate of the C&W attack, which can be viewed as a (very long) multi-patch attack. This encourages us that our multi-patch attacks, which impose a rigid structure on the attack can help design universal attacks. We leave this investigation to future work.

**ImageNet** ImageNet attacks are naturally harder for APSyn and require more patches, candidates, and optimization iterations compared to the other datasets we consider. Naturally, this affects the execution time. Figure 4 shows the success rate, $L_2$ and $L_0$ distortion and execution time for different combinations of number of candidates and patches. Results show that the success rate significantly increases with the increase in the number of patches and candidates. Interestingly, the $L_2$ distortion is similar in all the combinations, while the $L_0$ distortion decreases for more patches. This is because attacks with ten patches often have more overlaps between patches than attacks with five patches.

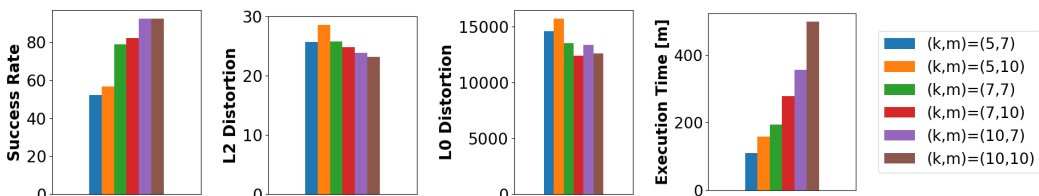

Figure 4: APSyn's success rate, $L_2$ and $L_0$ distortion, and execution time for ImageNet.

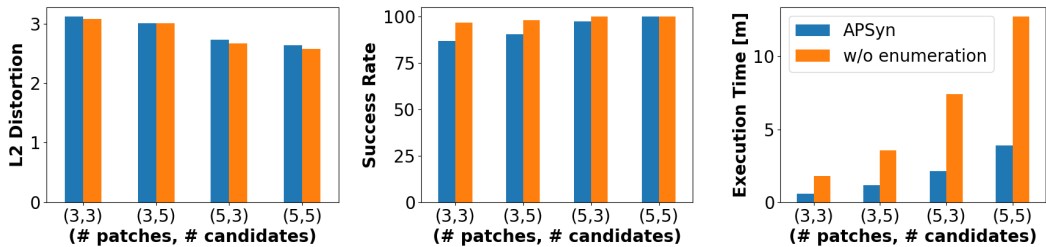

Figure 5: Comparison of enumerative synthesis vs. full-program optimization.

**Enumerative synthesis**  We next show the advantage of using enumerative synthesis over the more common approach in continuous optimization, which optimizes over all variables from the beginning. Figure 5 shows, for different numbers of patches and candidates $(k, m)$, the success rate, $L_2$ distortion, and execution time of both approaches. Results show that both approaches obtain similar success rate and $L_2$ distortion, however enumerative synthesis drastically reduces the execution time: up to 3x. This is significant because APSyn's execution time is not negligible.

**Examples of adversarial programs**  Lastly, Figure 6 illustrates some of the attacks that APSyn generates. For each image, the figure shows the adversarial examples of C&W and APSyn, and the differences between the adversarial examples and the input image. The pixels in the difference image range over $[0, 1]$ in order to show whether the perturbation brightened or darkened the pixel (i.e., 0.5 corresponds to an unperturbed pixel, 0 for maximal darkening, and 1 for maximal brightening). The examples illustrate that APSyn's attacks are imperceptible, even though the $L_2$ distortion is higher than C&W's. The examples also demonstrate that because our attacks are restricted to patches, they provide an explanation to the exploitable regions. For example, in the bag example, APSyn creates a patch which darkens the purse handle; in the digit six example, APSyn brightens the upper part using a horizonal patch and darkens the lower part of the digit towards making it look like the digit five; a similar pattern is observed in the digit nine example. We believe this opens a new direction of explaining network robustness using programs. We leave this to future work. Another interesting property is that patches tend to concentrate in and around the main object. This may suggest that our attacks can be directly executed on physical objects. Lastly, the examples show that APSyn sometimes returns programs that are shorter than the maximal number of patches $k$. This demonstrates the advantage of using enumerative synthesis which favors shorter solutions.

# 6  RELATED WORK

Our work is mainly related to Carlini & Wagner (2017) and Brown et al. (2017). Carlini & Wagner (2017) consider imperceptible attacks and propose several losses. While successful, it is hard to physically execute this attack because it perturbs a very large number of pixels. On the other hand, Brown et al. (2017) consider an attack executed by a visible patch. Their attack looks for the optimal perturbation, given the patch position and dimensions. This work also aims at generating patches universal to scenes and robust to transformations. In this work, we show that it is possible to enjoy both worlds by allowing multiple patches, each incurring a small $L_2$ distortion. Unlike Carlini & Wagner (2017), our attacks are defined over discrete variables. We show that by estimating the gradients of the discrete variables and by relaxing them to keep real values during optimization,

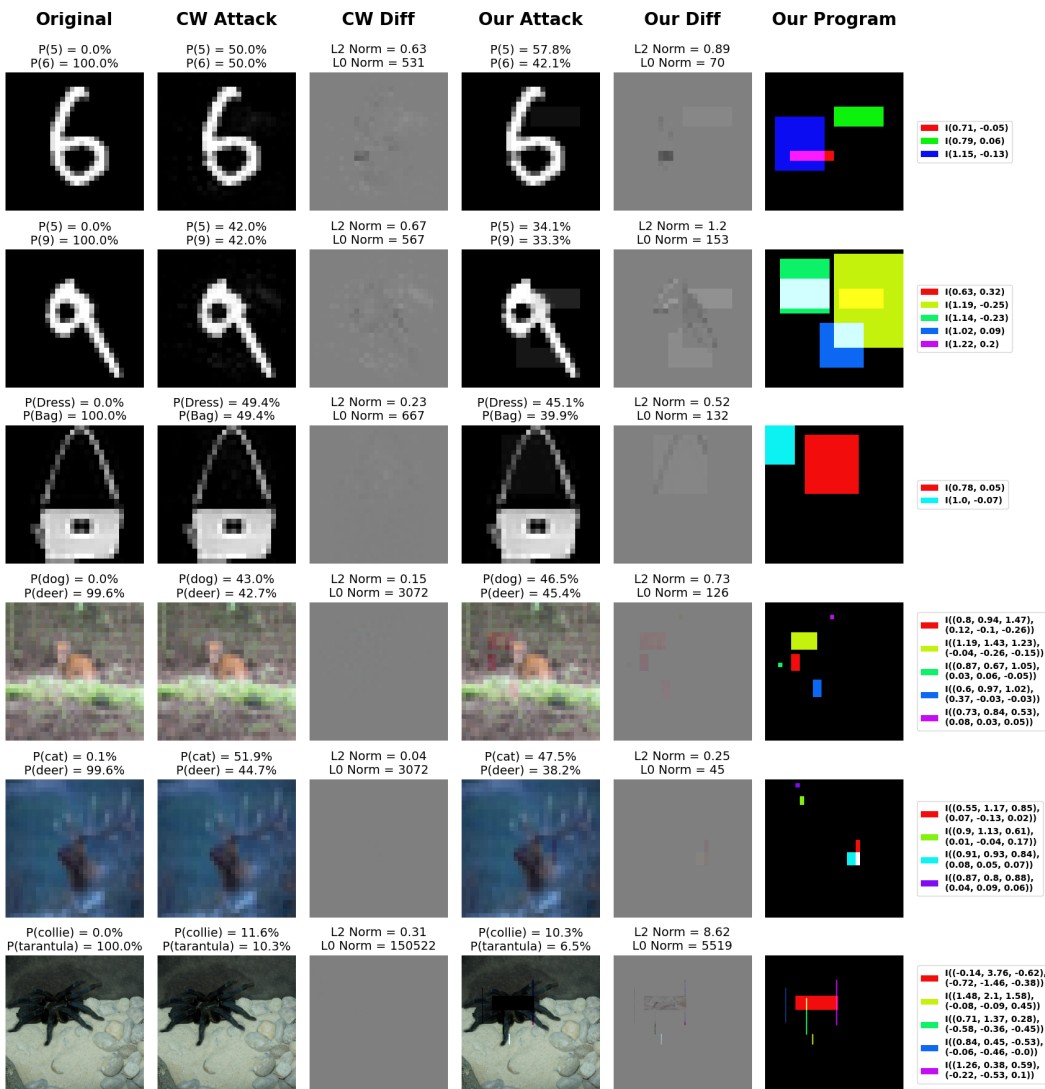

Figure 6: Examples of the attacks APSyn generated compared to the C&W attack.

the C&W loss is highly effective for producing adversarial examples even if the perturbation region is constrained. Unlike Brown et al. (2017), our work shows how to efficiently search for optimal position and dimensions of patches, which is crucial for minimizing distortion without reducing the success rate. We further build on program synthesis to search for attacks which minimize the number of patches. This allows APSyn to identify short and simple attacks and reduce execution times. Our attacks can be further optimized for universality by leveraging the ideas of Brown et al. (2017). We leave this for future work. Differently from both these works, our attacks have a rigid structure and restricted kind of perturbations. This forces APSyn to look for the most exploitable regions, which makes our attacks more explainable.

**Digital Attacks**    Since the introduction of adversarial examples (Szegedy et al., 2014; Goodfellow et al., 2015), many works have proposed adversarial attacks for various domains, perturbation kinds, and distortion metrics. A large body of works focuses on digital attacks for images, which induce artificially crafted additive noise to a digital image before using it as an input to a DNN. Some notable works include Carlini & Wagner (2017); Moosavi-Dezfooli et al. (2016); Modas et al. (2019); Papernot et al. (2016); Moosavi-Dezfooli et al. (2017); Madry et al. (2018). These works focus on minimizing $L_1$/ $L_2$/ $L_\infty$ distortion, which are differentiable functions, and are thus amenable to gradient-based optimization. Fewer works focus on minimizing $L_0$ distortion, which is a discrete

metric, and thus requires different kind of optimization. For example, Su et al. (2019) employ a differential evolution strategy. While successful for few pixels, our attempts to use it for multi-patch attacks suggest that this technique struggles to succeed when the number of perturbed pixels is large. Laidlaw & Feizi (2019) compute functional attacks, a concept which is similar in spirit to our patch programs. However, their attacks may perturb a large portion of the image pixels.

**Physical Attacks**  Digital attacks are generally harder to execute in a realistic setting where an attacker has to manipulate physical objects. This has led to studying physical-world attacks. Kurakin et al. (2017) use a cell-phone camera to photograph printed images of adversarial attacks and showed that they remain adversarial and are robust to the transformations and the noise resulting from the phone and camera processing. In contrast, Lu et al. (2017) use traffic sign images to show that adversarial examples are less successful when viewed from different angles and distances. Several works propose to use stickers and 3D-printed objects to execute adversarial attacks in a realistic environment. Sharif et al. (2018) propose a generative approach to generate 3D adversarial glasses to fool face recognition classifiers. Athalye et al. (2018) generate adversarial attacks using printed 2D images and 3D objects which are robust under synthetic image transformations. Eykholt et al. (2018) rely on specially-crafted stickers on physical traffic signs to generate adversarial attacks that are robust for different distances and angles. This attack computes multiple stickers, whose position and dimensions are determined by first running their optimization on the entire region, but favoring sparse perturbations. Based on the resulted attack, stickers are positioned in the most vulnerable regions, and then the optimization runs again. In contrast, APSyn relies on enumerative synthesis to guarantee it computes a minimal patch sequence. Li et al. (2019) generate robust adversarial attacks by placing stickers on camera lens. Jere et al. (2019) introduce scratch attacks for image captioning. Another approach to physical attacks assumes the adversary can change the network's behavior to respond to malicious objects (e.g., trojan attacks  Liu et al. (2018); Ji et al. (2018) and poisoning attacks (Gu et al., 2017)). Brown et al. (2017) introduce robust and universal attacks via adversarial patches. Karmon et al. (2018) propose smaller and less visible adversarial patches, but these are not universal. (Liu et al., 2019a; Bai et al., 2021) rely on GANs to generate inconspicuous patches. (Liu et al., 2019b; Lee & Kolter, 2019; Thys et al., 2019; Lang et al., 2021; Wu et al., 2020; Xu et al., 2020) use different variations of adversarial patches to fool object detection models causing them to miss the target object. Tu et al. (2020) generate printed 3D objects that fool LIDAR detectors. Liu et al. (2020) generate realistic patches on grocery items using a bias-based approach. Lovisotto et al. (2021) generate short-lived localized adversarial attacks by utilizing an RGB projector and modeling the three-way relationship between a surface, a projection and the camera-perceived image. In contrast, our attacks rely on optimization of patch sequences, which enable APSyn to reduce the $L_2$ distortion to an imperceptible level. A key factor in the success of APSyn is the optimization of the patches' dimensions and position. While we have not physically executed our attack, we believe it can be combined with the above attacks to reduce their distortion.

**Patch Attacks**  In addition to the above patch attacks which have been designed for a physical setting, several other works showed successful patch attacks. Andriushchenko et al. (2020) rely on random search to generate attacks consisting of squares. Unlike our work, their attacks consist of a very large number of patches and their $L_0$ distortion is very high. Yang et al. (2020) rely on reinforcement learning to generate a visible texture-based patch picked from a predetermined dictionary. Rao et al. (2020) generate visible patch attacks, and rely on a similar idea to ours to optimize the patch position. In contrast, our approach is to first to create minimal multi-patch attacks, where patches can overlap, and their position, dimensions, and perturbation factors are jointly optimized.

## 7 CONCLUSION

We presented APSyn, a synthesizer of multi-patch attacks. APSyn searches for a minimal sequence of patches that produces an adversarial example with a minimal $L_2$ distortion. Unlike previous patch/sticker attacks, APSyn optimizes over the patches' position and dimensions. To this end, we introduce a gradient estimate for the discrete variables and relax them to keep real values during the optimization. To further minimize the number of patches and improve execution times, APSyn employs enumerative synthesis. We evaluate APSyn on four datasets and show that it generates imperceptible attacks with few patches.

## ETHICS STATEMENT

Our work introduces a new kind of adversarial attacks and thus has a potential negative impact. However, since our approach can pinpoint more exploitable regions and generate attacks in the form of a short program, we believe it can help network designers understand vulnerability aspects of their networks. This can help them in improving the robustness of their network, e.g., using defenses.

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

Figure 7: Comparison of APSyn with a baseline that does not perform optimization and a baseline that does not perform adaptive optimization.

## A  APPENDIX

In this section, we study the effectiveness of our discrete optimization (via gradient estimation and adaptive optimization). To this end, we compare APSyn to two baselines. The first baseline is identical to APSyn but does not optimize the discrete variables. That is, patches are added one-by-one, their parameters are initialized as described in Section 4 (including our feature ablation approach for initializing patch position), and the continuous variables are optimized for the C&W loss with gradient descent. The second baseline is the same as APSyn, but without our adaptive optimization of the discrete variables (employing Hill Climbing and Adam steps). That is, the second baseline jointly optimizes all variables (discrete and continuous), and the step size of all discrete variables is the same. In this experiment, we set the maximal number of patches to $k = 5$, and the number of candidates to $m = 7$. Figure 7 shows the success rate, $L_2$ distortion, and $L_0$ distortion of all approaches. Results show that our discrete optimization enables APSyn to obtain a higher success rate and significantly lower $L_2$ and $L_0$ distortion. That is, it improves both the success rate and the imperceptibility of the generated attacks. Results also show that our adaptive discrete optimization enables APSyn to obtain a significantly lower $L_0$ distortion, and also improve success rate and $L_2$ distortion.

