# OpenReview forum: "Towards Physical, Imperceptible Adversarial Attacks via Adversarial Programs"
_ICLR.cc/2022/Conference — ICLR 2022 Submitted_

### Official Review · Reviewer_Uj1D · 2021-10-30

**Correctness:** 3
**Technical Novelty And Significance:** 2
**Empirical Novelty And Significance:** 2
**Recommendation:** 5
**Confidence:** 4

**Main Review:**

This work frames the adversarial attack as a program synthesis problem, which is interesting. However, I don't think this problem setup is very well-motivated. Specifically, there might be simpler adaptions of existing adversarial attack algorithms that can achieve similar or better performance, while they do not require a computationally expensive program search process. I discuss the details as follows.

1. I find that the confidences of the generated adversarial examples are all pretty low. Have you tried to generate adversarial examples that are highly confident in the target labels, e.g., >95%? What is the attack success rate of generating such strong adversarial examples, and what is the execution time? My concern is that this requirement could further slow down the program search process, and makes the proposed approach even much slower than the baseline adversarial attack algorithms.

2. Is it necessary to search for patch positions? If the goal is to generate multi-patch adversarial examples, one simpler approach is to randomly generate several masks, and optimize the perturbations in the regions covered in the mask. Have you run this baseline?

3. Is there any reason to build upon C&W, instead of some other adversarial attack algorithms, e.g., PGD? Have you tried existing L0 adversarial attacks, and does your algorithm perform better than these existing attacks?

4. The authors demonstrate some generalization results, i.e., using the same program to generate perturbations for multiple images. While it is interesting to see some transferability, the current generalization rate is still pretty low, and is not competitive compared to prior works focusing on universal adversarial perturbations. Given that the approach still needs to generate one program per input image, in general the proposed approach is computationally expensive, and thus its usage is unclear.

**Summary Of The Paper:**

This paper studies multi-patch adversarial attacks, where each perturbation is generated via a program. Since the program includes both discrete variables and continuous parameters, they design a gradient estimation scheme for optimizing the parameters, and a program search algorithm to search for programs with increasing lengths. On several image classification benchmarks, they demonstrate that their approach achieves decent attack success rates and low L2 distortion, which are comparable to the C&W attack. Meanwhile, their attack requires much lower L0 distortion.

**Summary Of The Review:**

This work frames the adversarial attack as a program synthesis problem, which is interesting. However, I don't think this problem setup is very well-motivated. Specifically, there might be simpler adaptions of existing adversarial attack algorithms that can achieve similar or better performance, while they do not require a computationally expensive program search process.

---

> ### Author Response · Authors · 2021-11-14
> **Author Response**
>
> We thank the reviewer for the comments. We next address the comments.
>
> 1. In the vast majority of adversarial attack settings, the goal is to create a successful attack without constraining the confidence to be above some threshold. Naturally, there is a trade-off between the attack’s confidence, the execution time, and the distortion level. This is true for any adversarial attack. Nevertheless, our framework can easily be extended to allow the user to add a confidence constraint.
>
> 2. Yes, we tried it and this approach has a significantly lower success rate, even when we allow it longer execution times. The trajectory-based approach we introduce is more efficient for images than pure random search methods because we can utilize the spatial locality of images.
>
> 3. C&W propose a very successful loss, and other works also build on it (e.g., Bai et al., “Inconspicuous adversarial patches for fooling image recognition systems on mobile devices”). Our framework can easily be extended to support other attacks’ losses (as long as these are differentiable). Our attacks balance L2 distortion and L0 distortion and thus will not perform better in either metric compared to attacks which optimize only one of the norms. However, we show that this balance allows us to compute imperceptible multi-patch attacks, where patches are parameterized by a single scale and shift factors and patches can overlap, which is a new setting of adversarial examples.
>
> 4. The goal of measuring the generalization rate is not to claim that our work provides universal attacks. It is not realistic to expect a very high generalization rate when we do not compute our attacks towards obtaining this goal. However, the higher generalization rate does exemplify that our simple attacks generalize better than the C&W attack, on which we build. This shows that simplicity (i.e., constraining the number of degrees of freedom) increases generalizability. In other words, our structured multi-patch attacks transfer better to other inputs. The goal of computing a program in our setting (i.e., the patch sequence) is to provide a simple attack, which is easier to physically execute (like was demonstrated by previous works on patch attacks).

---

> > ### Comment · Reviewer_Uj1D · 2021-11-30
> > **After Author Response**
> >
> > I thank the authors for the response and the revision. However, I still don't think the program synthesis formulation is well-motivated, and the authors did not address the concerns on scalability. Therefore, I keep my initial score.

---

> > > ### Author Response · Authors · 2021-11-30
> > > **Response to After Author Response**
> > >
> > > Thank you for your feedback. We apologize for not addressing your concerns about our program synthesis formulation and scalability in our previous response. We thought the numbered points detail all the reviewer’s concerns. Here is our response:
> > > - Program synthesis formulation: Our algorithm draws inspiration from program synthesis algorithms because it enumerates programs of increasing size and at each step employs a customized search procedure (numerical optimization).  A common property of program synthesizers, which is highly relevant to us and is the reason why we phrased our problem as a program synthesis task, is that they look for a shortest solution. This is in contrast to other optimization problems (e.g., standard L2 attacks) which do not view the number of instructions/unique perturbations as an optimization goal. That said, we are willing to tone down the formulation as a program synthesis task if all reviewers believe it is better.
> > > - Scalability: Our attack’s complexity is polynomial in the input size (we can include a complexity analysis). The execution times naturally increase for deep networks and high-resolution images (similarly to training).

---

### Official Review · Reviewer_xXLy · 2021-11-01

**Correctness:** 4
**Technical Novelty And Significance:** 2
**Empirical Novelty And Significance:** 2
**Recommendation:** 6
**Confidence:** 5

**Main Review:**

The idea of the paper is well motivated, and it is a natural direction to explore the possibility of imperceptible patches. The formulation of adversarial programs is neat and quite clever to get around the discrete nature of gradient. I do notice a couple of places where the reality is less optimal than the assumptions such as eta_c vs eta_d, these empirical values are sometimes physical constraints that researchers have to deal with.
In general, I like the evaluations shown in figure 2-figure 5, these results indicate the proposed method is effective.


**Summary Of The Paper:**

This paper focuses on crafting imperceptible patches to fool image classifiers. The method leverages program synthesis and numerical optimization  and produces perturbations with a lot less perturbed pixels compared to baselines. The experiments on various image datasets demonstrate the effectiveness of the method.

**Summary Of The Review:**

However, I have 2 minor concerns:
1. The title of the paper boasts “physical”, and yet figure 6 indicates the experiments were mainly carried out digitally. What’s the real physical performance? I.e. how effective is the proposed method when deployed in the real physical world?
2. This work mainly used l2 distance to measure distortion, but l2 distance sometimes could be flawed itself, how about measure the distance between 2 distributions? Would that change the results?

---

> ### Author Response · Authors · 2021-11-14
> **Author Response**
>
> We thank the reviewer for the comments. We next address the comments.
>
> 1. Our work considers multi-patch attacks. Patch attacks have been shown by numerous works to be easier to physically execute than attacks that perturb the entire image. Executing a physical attack has orthogonal challenges (e.g., tuning the camera).
> This is the reason why our paper is titled “*towards* physical, imperceptible adversarial attacks via adversarial programs”. We believe our novel setting of multi-patch attacks, where patches are parameterized by a single scale and shift factors and patches can overlap, is interesting. We also believe our work opens a new direction of explaining network behavior and robustness using adversarial programs.
>
> 2. We are not sure which distributions the reviewer is referring to, and we would appreciate it if the reviewer can elaborate on this point and perhaps even provide citations to papers that have done this comparison.

---

### Official Review · Reviewer_tecw · 2021-11-01

**Correctness:** 3
**Technical Novelty And Significance:** 2
**Empirical Novelty And Significance:** 1
**Recommendation:** 3
**Confidence:** 5

**Main Review:**

Strengths:

(I) The paper proposes an approach for optimizing the patch location/dimension (while prior works usually fix the patch location/dimension for simplicity).

(II) It is interesting to see the authors formulating the attack as a program synthesis problem (though I have concerns about the necessity of such a formulation; see weakness II below).

Weaknesses:

(I) The paper does not provide enough motivation and justification for its problem setup. It might be trivial to succeed in this attack
1. The problem formulation does not have any constraint for the L_0 distortion (i.e., the sizes and the number of patches)! Therefore, the problem reduces to the global L_2 perturbation (e.g., the C&W attack) when the patch covers the entire image. As a result, there are existing solutions (e.g., functional adversarial attacks [1]) to this problem, which undermines the contributions of this paper. I would suggest the authors have both L_0 and L_2 constraints in the problem formulation and algorithm design. Also, I would appreciate it if the authors could clearly state the challenges of this problem, and discuss the problem/algorithm with relevant works [4,5] in more detail.

2. The attack formulation does not fully capture the nature of physical attacks. The paper assumes that the patch attack is always physically realizable. This is not true; the perturbations within the patch can be too subtle/small to be realized (e.g., printed) in the physical world (Sharif et al. [2] defined the non-printability score). Since the paper does not evaluate the attack in the physical world setting, I would suggest the authors rephrase the problem as finding a balance between L_2 and L_0 attacks (instead of claiming the attack is physically realizable).

3. The paper does not explain why it only considers pixel scaling and shifting within the patch. There can be a more imperceptible and more effective perturbation (compared to scaling and shifting) for the patch. Maybe the authors assume that scaling and shifting make the attack more physically realizable, but more justifications are necessary.

4. The authors formulate the attack as a program synthesis problem; however, I find the core idea does not involve anything about problem synthesis. The core algorithm is to (1) initialize a random patch; (2) optimize over the pixel values and (discrete) patch locations; (3) depending on a certain condition, continue to add another patch or return the generated adversarial examples. The authors did not use any technique from program synthesis in their algorithm; therefore, I find such a formulation only adds unnecessary complexity to the paper, and I would suggest discarding this formulation.

(II) The proposed method lacks novelty

1. As discussed in weakness I.4, the authors do not adapt any technique from program synthesis in the proposed approach, though they formulate the problem as program synthesis.

2. The problem of imperceptible patches is not new [4,5].

3. The loss for the attack is from existing work (C&W attack).

4. The estimation of gradients of discrete values looks straightforward; the evaluation (Figure 5) shows that this approach only improves the attack efficiency at the cost of a small drop in attack success rate. Also, the experiment does not have enough comparison with baselines (see weakness III) to demonstrate the effectiveness of the gradient estimation.

(III) The experiment lacks a reasonable baseline comparison and a comprehensive ablation study.

1. The attack performance comparison seems unfair. The paper compares APSyn with C&W, while C&W is designed for global perturbations. Therefore, it is unsurprising to see APSyn has a smaller L_0 distortion and a large L_2 perturbation(Figure 3). Moreover, there is a relevant paper that discusses patch location optimization [3]. Also, imperceptible patches are studied in prior works [4,5]. These papers should be a more reasonable baseline to compare with.

2. There is another simple baseline for this approach: randomly pick the patch location/dimension and then use the C&W attack with an L_2 constraint to find the perturbations. This should be a baseline to show that the discrete value optimization for patch location/dimension is useful.

3. Why did the experiment consider all 9 target classes for the 10-class CIFAR-10, MNIST, Fashion-MNIST datasets but only consider 3 target classes for ImageNet (Note that the authors consider 10 classes for ImageNet but only 3 *target* classes)? Is it because your approach is expensive for high-resolution images? Moreover, the experiment only considers *three* images for each dataset; its results may not be representative.

4. For the experiment setup, the authors simply report the default hyperparameters (e.g., \lambda, \eta, scale, mean/std of Gaussian) without any justification. There is no ablation study. I suggest the author at least report the trade-off between attack success rate and attack time (balanced by the number of patches/candidates).

(IV) Finally, though it is minor, the mathematical presentation can be improved. Here is an incomplete list of notation issues that I noticed:

1. $m$ is used multiple times (as the mask and also the number of patch candidates)
2. $\lambda$ is used multiple times (in the formulation of adversarial programs, and in C&W attack as the balancing weight)
3. In “adversarial programs”, there is an inconsistency between $I_{\alpha,\beta,p}$ and $I_{\alpha,\beta,m}$.

[1] Laidlaw et al., “Functional Adversarial Attacks”, NeurIPS 2019.

[2] Sharif et al., “Accessorize to a Crime: Real and Stealthy Attacks on State-of-the-Art Face Recognition”, CCS 2016.

[3] Rao et al., “Adversarial Training against Location-Optimized Adversarial Patches”, ECCV  workshop 2020.

[4] Liu et al., “Perceptualsensitive GAN for generating adversarial patches”, AAAI 2019.

[5] Bai et al., “Inconspicuous adversarial patches for fooling image recognition systems on mobile devices”.


**Summary Of The Paper:**

This paper proposes APSyn to generate a sequence of imperceptible adversarial patches for a physical and imperceptible attack against image classification models. This attack is a combination of physical patch attacks and conventional imperceptible perturbation attacks. In APSyn, the authors formulate the attack problem as adversarial program synthesis and use gradient-based optimization to search for the optimal solution (a method for estimating the gradient for discrete values is also proposed). The experiments on small subsets of the MNIST, Fashion-MNIST, CIFAR-10, and ImageNet demonstrate that the proposed approach can achieve smaller L_0 distortion and larger L_2 distortion compared to the C&W attack.

**Summary Of The Review:**

Based on the three issues listed below, I would recommend rejecting this paper.

1. The problem is not well motivated and justified (see weakness I), and it might be trivial to succeed in this attack (see weakness III.2). I would suggest reformulating the problem setup and algorithm (see the main review for detailed suggestions) and clearly stating the challenges of this problem.

2. The proposed solution seems to be a very straightforward combination of several existing algorithms. The simplicity of the algorithm is not bad; however, I would encourage the authors to clearly state (1) why this problem is challenging; (2) why their proposed approach is non-trivial and novel; (3) how effective the proposed approach is (requires more baseline comparisons).

3. The evaluation is incomprehensive, and the performance comparison chooses an unreasonable baseline (see weakness III). I would suggest adding more experimental comparisons with other baselines to demonstrate the effectiveness of the proposed approach (see the main review for detailed suggestions).

---

> ### Author Response · Authors · 2021-11-14
> **Author Response - (I) + (II)**
>
> We thank the reviewer for the comments. We will include the suggested related work. We next address the reviewer’s comments.
>
> (I)
>
> 1. Limit on the L0-norm: we limit the number of patches to a predefined number k and each patch has a single scale and shift factors. Thus, our problem does *not* reduce to the global L2 perturbation and [1] does not solve our problem because to implement it with our patch instructions we would require a patch per pixel, which will exceed the k patch budget. Note that in our experiments we pick k to be very small (up to k=10; for MNIST, CIFAR-10, and FashionMNIST, k=5 is usually sufficient). Additionally, our algorithm looks for the smallest k’ (<=k) for which APSyn returns an adversarial program. To optimize for the smallest k’, we take the enumerative synthesis approach and search for programs of increasing size. We explain the challenges both in the introduction and at the end of Section 2, but we will extend these parts and explain them better. Regarding the mentioned works:
> [4] focuses on visible patches and assumes a fixed number of patches. They also do not optimize the dimensions of the patches.
> [5] uses a single patch with predetermined position and dimensions that cannot adapt and be optimized based on the perturbation values. Moreover, in their approach every pixel within the patch is perturbed independently, while in ours each patch perturbs its pixels with the same parameters. Therefore, our approach produces imperceptible attacks which are simpler, more explainable, and with significantly fewer degrees of freedom. We also allow patches to overlap and create higher capacity attacks with complex patterns while maintaining the simple formulation and a minimal number of degrees of freedom.
>
> 2. Our work considers multi-patch attacks. Patch attacks have been shown by numerous works to be easier to physically execute than attacks that perturb the entire image. Executing a physical attack has orthogonal challenges (e.g., tuning the camera).
> This is the reason why our paper is titled “*towards* physical, imperceptible adversarial attacks via adversarial programs”. We believe our novel setting of multi-patch attacks, where patches are parameterized by a single scale and shift factors and patches can overlap, is interesting. We also believe our work opens a new direction of explaining network behavior and robustness using adversarial programs.
>
> 3. We believe that scaling and shifting perturbations are easier to physically execute, but APSyn is easily extensible to any differentiable perturbation. If the reviewer has examples of other useful perturbations, we are happy to include experiments with these perturbations.
>
> 4. Enumerative synthesis is a simple synthesis approach which considers programs of increasing size with the goal of finding a shortest (and simplest) solution. The challenge is often to rank programs as the program’s complexity increases (see Program Synthesis, Gulawani et al., Chapter 4 https://www.nowpublishers.com/article/Details/PGL-010). Unlike many program synthesis tasks, our instruction set is very large. Thus, a simple ranking is futile. Instead, we rely on numerical optimization. Since our instructions involve discrete parameters, we show how to estimate their gradients. Our combination of program synthesis and numerical optimization shows how to build on each technique's strength to compute successful, short multi-patch attacks.
>
> (II)
>
> 1. Program synthesis novelty: we combine enumerative synthesis with numerical optimization (including adaptation for the discrete variables). As we explain at the end of Section 2, standard enumerative synthesis is futile because the space of instructions is too large. Thus, standard search approaches (e.g., instruction ranking) is futile.
>
> 2. The problem of imperceptible multi-patch attacks is new. [4] focuses on visible patches, whereas [5] focuses on a single patch, with predetermined position and dimensions that cannot adapt and be optimized based on the perturbation values. Moreover, in their approach every pixel within the patch is perturbed independently, while in ours each patch perturbs its pixels with the same parameters. Therefore, our approach produces imperceptible attacks which are simpler, more explainable, and with significantly fewer degrees of freedom.
>
> 3. We have not claimed that the novelty of our work is a new loss function (incidentally, [5] also rely on the C&W loss).
>
> 4. Indeed, our estimation of the discrete variables’ gradients is not complicated. We see this as a strength of our approach that such estimation is sufficient to efficiently search in the large and complex space of multi-patch attacks, where the patches’ dimensions, positions, and perturbations are optimized throughout the search procedure (which is a novel setting).

---

> > ### Author Response · Authors · 2021-11-14
> > **Author Response - (III) + (IV)**
> >
> > (III)
> >
> > 1. As the reviewer points out, our attack balances L0 and L2 distortion. Since our loss relies on the C&W loss, it is the closest baseline to our setting. In fact, our attack can be seen as the C&W attack with multi-patch constraints. We are happy to include more comparisons but [3] focuses on a single visible patch, which is a different setting than ours. Additionally, [3] does not optimize the patch dimensions, in contrast to us. We also optimize multiple patch parameters jointly, which is irrelevant to their setting. Thus, our approach is more general. [4] does not consider imperceptible attacks and thus it is not a suitable baseline.  [5] focuses on a single patch, with predetermined position and dimensions that are not optimized based on the perturbation values. Also, every pixel within the patch is perturbed independently, while in ours each patch perturbs its pixels with the same parameters. Thus, to compare, we should restrict our attack to a single patch and allow our patches to take any value independently. We are happy to include this comparison if the reviewer is interested, but we are not sure what insight this comparison will provide.
> >
> > 2. During the design process of APSyn, we compared to the random baseline the reviewer describes, and we observed APSyn is significantly better. We are happy to include the results of this comparison. Due to lack of space, we will include these experiments in the appendix.
> >
> > 3. Yes, but this is expected (similarly to training, which is more expensive for high-resolution images): our algorithm’s complexity is polynomial in the input size. That said, we have seen similar results for attacks for other source and target classes, and thus our results are representative. We are happy to consider more experiments if the reviewer believes it is necessary.
> >
> > 4. We performed ablation testing for determining the hyperparameters values. Due to space restrictions and because we believe the experiments we included are more insightful, we have not reported these results. The trade-off between the success rate and execution time (balanced by the number of patches/candidates) is reported in Figure 3, 4, and 5.
> >
> > (IV)
> > We will fix the notation issues.

---

> ### Comment · Reviewer_tecw · 2021-11-14
> **Reviewer Responses to Author Rebuttal**
>
> My responses to the authors’ rebuttal are provided below. Overall, the rebuttal does not demonstrate any major misunderstanding in my initial review comments. Therefore, I will keep my score unchanged.
>
> (I)
> 1. In your formulation, you do not constrain **the size of each patch**. Your optimization over the patch dimension does not have size constraints as well (feel free to correct me if I am wrong). Therefore, your formulation allows for an extremely large patch to cover the entire image. That is why I believe the formulation is problematic.
> 2. Fair. Maybe emphasize the word “toward” a bit more in the paper?
> 3. My comment is: why are scaling and shifting easier to physically execute? It is the authors’ job to justify their problem setup. Neither the paper nor the rebuttal addresses this question.
> 4. The response does not address my concern. As discussed in my initial comments, the entire algorithm can work without any concept of program synthesis. Introducing program synthesis only adds unnecessary complexity.
>
> (II)
> 1. see (I).4. above
> 2. Moving from one-patch to multiple-patch is not a novelty! Instead, you give the attacker more flexibility, and thus make the attack easier.
> 3. Sure.
> 4. The rebuttal does not respond to my second part of the comment: the effectiveness of the proposed approach. A good approach can be simple but must be effective.
>
> (III)
> 1. If the authors find it unfair to compare with prior works. There should be at least enough ablation study of different modules of the attack design. Comparison with prior works (or ablation study for each attack module) can demonstrate the contribution of each of your design components (e.g., using multiple patches, using scaling and shifting, optimizing patch dimension) and help readers better understand and evaluate the performance of the proposed algorithm.
>
> For example, if you find your algorithm cannot work for a single patch, it means that your attack is not as effective as prior works. Your attack succeeds merely because you allow the attacker to corrupt more patches/pixels.
>
> Another example, if you find optimizing patch dimension has a similar performance as fixing the patch dimension. It means that your proposed method is not effective.
>
> Moreover, your comparison with C&W shows that you have a smaller L_0 but a larger L_2 perturbation. It is even hard to argue that your attack algorithm has non-trivial achievements.
>
> 2. This is a super important baseline and ablation study (I discussed the reason in the point above). Since there is no major revision process in ICLR. I would suggest the author provide the results during the rebuttal.
> 3. The attack complexity should be clarified in the paper. Even polynomial complexity could make the system unusable (224x224 images will be 49x more expensive than 32x32 images; for example, let’s think of 1hr vs 49 hrs).
> 4. My apologies for the comments about the trade-off between success rate and execution time. A possible improvement could be adding captions for sub-figures in these large figures. A minor comment: for ICLR submission, “Authors may use as many pages of appendices (after the bibliography) as they wish”.

---

> > ### Author Response · Authors · 2021-11-15
> > **Author Response**
> >
> > (I)
> >
> > 1. While we can constrain the size of the patches, it is not needed because we optimize the L2 distortion and we allow up to k patches, each with a single scale and shift factors. Existing global L2 perturbations will either exceed the number of patches allowed (we consider up to k<=10) or will result in very visible attacks. For example, the reviewer says that [1] and [5] solve our problem, however to express these attacks as a patch sequence in our language, they will require more than k>10 patches (actually, at least a few dozens), whereas we show our attack has high success rate and imperceptible distortion with at most 10 patches (often fewer). Constraining the size of patches may be interesting to allow an attacker to pose additional constraints on the attack.
> >
> > 2. We mentioned it in several sections, but we will emphasize it better.
> >
> > 3. The main reason we focus on scale and shift factors is because they are simple functions and easy to understand. To convince the reviewers that these are useful for our problem setup, we show our experiments. We have not claimed in the paper that scale and shift patches are easier to physically execute than other patches, and this is why we do not need to justify it. We merely shared with the reviewer our gut feeling, but no scientific claim was made. We apologize for that.
> >
> > 4. Our algorithm does in fact draw inspiration from program synthesis algorithms: it enumerates programs of increasing size and at each step employs a customized search procedure (numerical optimization). We are willing to tone down the formulation as a program synthesis task if all reviewers believe it is better.
> >
> > (II)
> >
> > 2. Please note that our patches are highly-constrained compared to prior patch attacks: our patches cannot perturb each pixel independently, but rather each patch is parameterized by a single shift and scale factors. Additionally, to the best of our knowledge, we are the first to consider multi-patch attacks where patches are allowed to overlap, and at the same time the patches’ dimensions, position, and perturbation factors are not fixed.
> >
> > 4. We have responded about measuring the effectiveness of the gradient estimation in (III) 1+2 (to which the reviewer has referred in the review). If there is an additional comment that we have missed, we would be happy to respond.
> >
> > (III)
> >
> > 1. We have not included ablation study due to space restrictions. We will include it in an extended version of the paper. Our patches are different from prior patch attacks (see II-2), thus even if we fail with one patch, it does not mean we perform worse from other attacks or that we perturb more pixels (because our patches can be smaller than the fixed-sized patches that other works consider). We will add an experiment to show the advantage of varying patches’ size and position over fixed patches.
> > We cannot expect to obtain a better L2 distortion than C&W if we use the same loss but with additional constraints. Nevertheless, our multi-patch attacks consider simple patches and we show we obtain a high success rate and imperceptible distortion, with a handful of patches.
> >
> > 2. We will include this experiment. As a side note, even if there is no formal major revision process, if the reviewers decide to accept this work, we will incorporate their suggestions.
> >
> > 3. Our attack’s complexity is similar to other attacks (we can include a complexity analysis). The execution time of APSyn is reported separately for each dataset, so the reviewer can see the execution time for ImageNet (it is not around 49 hours).
> >
> > 4. We will include sub-captions.

---

### Official Review · Reviewer_3vSm · 2021-11-02

**Correctness:** 3
**Technical Novelty And Significance:** 2
**Empirical Novelty And Significance:** 1
**Recommendation:** 3
**Confidence:** 4

**Main Review:**

1. This work is not novel as shape-based attacks are well-explored in the black-box setting [1-3]. The method proposed in this paper is very similar to Square Attack proposed in [1], except that it is done under the white-box setting rather than the original black-box. The related works section should mention and differentiate this paper from them.
2. The evaluation is lacking in terms of compared attacks. How does it compare to the popular projected gradient descent (PGD) (originally introduced in [4]), and generally stronger white-box attacks such as [5-7]? The results are missing important comparisons here.
3. The reviewer believes that introducing the patch-based attack as an application of SyGuS is detrimental to the contribution of the paper, for the reasons below:
	1. The adversarial programs are not actual computer programs as they are not Turing complete. They can simply be reduced to sequences of patches, or even sets of patches if order independent.
	2. SyGuS is not rigorously introduced in the paper, even the section introducing APSyn does not mention SyGuS and how APSyn is related. It is unclear why the authors chose to emphasize the patches as programs.

Some other minor issues:
* In the abstract, it was mentioned that “the average L2 distortion is greater than the average L2 distortion of the C&W attack by up to 1.2.” The reviewer is confused how a greater L2 distortion than the baseline is better. Perhaps you mean smaller?

References:
```
[1]: Andriushchenko et al., Square Attack: a query-efficient black-box adversarial attack via random search, ECCV 2020. https://arxiv.org/abs/1912.00049
[2]: Jere et al., Scratch that! An Evolution-based Adversarial Attack against Neural Networks. https://arxiv.org/pdf/1912.02316.pdf
[3]: Yang et al., PatchAttack: A Black-box Texture-based Attack with Reinforcement Learning, ECCV 2020. https://link.springer.com/chapter/10.1007/978-3-030-58574-7_41
[4]: Mary et al., Towards Deep Learning Models Resistant to Adversarial Attacks. https://arxiv.org/abs/1706.06083
[5]: Croce et al., Reliable evaluation of adversarial robustness with an ensemble of diverse parameter-free attacks. ICML 2020. http://proceedings.mlr.press/v119/croce20b.html
[6]: Yu et al., LAFEAT: Piercing Through Adversarial Defenses with Latent Features. CVPR 2021. https://openaccess.thecvf.com/content/CVPR2021/papers/Yu_LAFEAT_Piercing_Through_Adversarial_Defenses_With_Latent_Features_CVPR_2021_paper.pdf
[7]: Tashiro et al., Diversity Can Be Transferred: Output Diversification for White- and Black-box Attacks. NeurIPS 2020.
```

**Summary Of The Paper:**

This paper proposes a white-box attack to generate rectangular patches of pixel color shifts as a means to generate adversarial examples for DNNs. As direct gradients for discrete parameters $w$ in a patch is not obtainable, the gradients are approximated with differences such as $L(w + 1) - L(w)$.


**Summary Of The Review:**

The reviewer cannot recommend acceptance because this paper is lacking in novelty and empirical evaluation.

---

> ### Author Response · Authors · 2021-11-14
> **Author Response**
>
> We thank the reviewer for the comments. We will include the suggested related work. We next address the reviewer’s comments.
> 1. We have not claimed that our novelty lies in the shape-based attacks (see our Related Work section). Our novelty is in computing imperceptible multi-patch attacks using a handful of patches, which are easier to physically execute. Moreover, we also allow patches to overlap and create higher capacity attacks with complex patterns while maintaining the simple formulation and a minimal number of degrees of freedom.
> [1] is different both in its goal as well as its approach: this work does not aim to minimize the number of squares or the perturbation amount (it is only restricted by a given threshold). Executing this attack requires a very large number of patches, which is harder than our attacks that use a handful of patches. Additionally, they rely on random search, whereas we combine enumerative synthesis with gradient descent. This requires us to define “gradients'' for the discrete variables. Using this approach is what enables us to obtain imperceptible perturbation using very few patches. Random search in our approach will result in visible attacks.
> [2] is not a patch-based attack and thus has different goals than ours. It also considers perceptible attacks and it is not easy to physically execute (unlike patches which have been shown to be easier to physically execute by numerous works). They also compute attacks using differential evolution. We experimented with differential evolution, for our setting, and observed it is inferior to gradient descent approaches.
> [3] generates perceptible attacks using a single patch whose perturbation is very visible and picked out of a predetermined dictionary. They also rely on an RL agent. In contrast, we focus on imperceptible attacks using a handful of patches, whose perturbations, dimensions and positions are jointly optimized using gradient descent. Moreover, our patches can overlap and thereby create complex patterns to reduce the overall perturbation, unlike the approach in [3].
>
> 2.
> We are happy to compare our attack with other attacks but we don’t believe these results will provide additional insight than the comparison with C&W. Our work aims to balance L2 and L0 distortions in order to generate simple attacks. Posing constraints to L2 attacks (via our patches) naturally requires an increase of the L2 norm. Thus, we merely aim at obtaining imperceptible attacks and not necessarily ones with lower L2 distortion than imperceptible attacks that can change all pixels independently (e.g., like PGD or [5-7]). We use the comparison to C&W to demonstrate that while our distortion is higher, it is not significantly higher. (This also answers the reviewer’s last comment: we meant higher in the abstract, we cannot expect to obtain a better L2 distortion than C&W if we use the same loss but with additional constraints).
>
> 3.
> Many works in program synthesis focus on DSLs which do not consist of standard program instructions. See for example:
>
> 1) [1] Synthesizing Geometry Constructions, Gulwani et al.  http://www.csl.sri.com/users/tiwari/papers/pldi2011-geometry.pdf
>
> 2) [2] SPREADSHEETCODER: Formula Prediction from Semi-structured Context, Chen et al., https://arxiv.org/pdf/2106.15339.pdf
>
> 3) [3] Automated Feedback Generation for Introductory Programming Assignments, Singh et al., https://www.microsoft.com/en-us/research/wp-content/uploads/2013/07/pldi13.pdf
>
> Practical program synthesizers typically do not address Turing-complete languages. A common property of program synthesizers, which is highly relevant to us and is the reason why we phrased our problem as a program synthesis task, is that they look for a shortest solution. This is in contrast to other optimization problems (e.g., standard L2 attacks) which do not view the number of instructions/unique perturbations as an optimization goal. We will improve the presentation to explain this point better and, in particular, explain why our setting falls into the SyGuS setting.

---

### Official Review · Reviewer_H5Ak · 2021-11-03

**Correctness:** 3
**Technical Novelty And Significance:** 3
**Empirical Novelty And Significance:** 3
**Recommendation:** 6
**Confidence:** 4

**Main Review:**

**Strengths**
1. This paper is easy to follow and well organized.
2. The proposed Adversarial Program Synthesizer (APSYN) is novel. This paper regards the generation of adversarial examples as a programming problem and designs an adversarial patch-based programming language to handle this issue. Experiments demonstrate the effectiveness of APSYN.
3. Considering that each adversarial patch only scales or shifts a specific part of the input, the generated adversarial examples are more explainable compared to other white/black-box gradients-based attack algorithms.

**Weakness**
1. This paper aims to generate a kind of physical imperceptible patches. However, there is no evidence of the effect of placing these patches into the physical world. These patches are valid and imperceptible in the constraint of $\ell_0$ or $\ell_2$ norm but that doesn’t mean they are still valid in the physical world.
2. Based on Section 3 and Section 4, the proposed APSYN algorithm utilizes zero-order optimization to update its weights. So it seems like the APSYN is a black-box attack algorithm. However, in Section 2 this paper "assumes white-box access to the classifier" and there are no actual gradients appearing in the algorithm description which is confusing.
3. In terms of L2 distortion, the generated adversarial samples are larger than samples created by other attack algorithms (e.g., C&W).
4. Missing some related physical world patch attack methods in the related work [1,2].

[1] Wu, Zuxuan, et al. "Making an invisibility cloak: Real world adversarial attacks on object detectors." European Conference on Computer Vision. Springer, Cham, 2020.

[2] Xu, Kaidi, et al. "Adversarial t-shirt! evading person detectors in a physical world." European Conference on Computer Vision. Springer, Cham, 2020.

**Summary Of The Paper:**

The paper proposed a new adversarial perturbation by stacking imperceptible adversarial patches. Each adversarial patch could be regarded as a disturbance rectangle mask and contains learnable coordinates, scaling, and shifting factors. By applying an adversarial patch into the input sample (e.g., image), the values of input inside the mask will be scaled and shifted. These adversarial patches are stacked and generated by solving an adversarial program problem with zero-order optimization and numerical optimization.

**Summary Of The Review:**

Generally, this paper proposed a novel programming-based adversarial perturbation and it is effective and competitive in various benchmarks. However, this paper aims to generate a physical imperceptible perturbation but doesn’t well-supported by experiential results.

---

> ### Author Response · Authors · 2021-11-14
> **Author Response**
>
> We thank the reviewer for the comments, we will include the related work suggested.
> We next address the reviewer’s comments:
>
> 1. Our work considers multi-patch attacks. Patch attacks have been shown by numerous works to be easier to physically execute than attacks that perturb the entire image. Executing a physical attack has orthogonal challenges (e.g., tuning the camera).
> This is the reason why our paper is titled “*towards* physical, imperceptible adversarial attacks via adversarial programs”. We believe our novel setting of multi-patch attacks, where patches are parameterized by a single scale and shift factors and patches can overlap, is interesting. We also believe our work opens a new direction of explaining network behavior and robustness using adversarial programs.
>
> 2. Our attack’s discrete parameters can assume a black-box access because we estimate gradients using zeroth order optimization, but the gradients of our continuous parameters are computed as usual and thus we assume a white-box access.
>
> 3. Our work aims to balance L2 and L0 norms in order to generate simple attacks. Posing constraints to L2 attacks (i.e., our patches) naturally requires an increase of the L2 norm. Note that C&W has a very large number of degrees of freedom  -- pixels can take any value in the valid range and there are no dependencies between perturbation of different pixels. In contrast, we constrain to up to k patches, each perturbs its pixels with the same parameters. Despite these constraints, we show that we still produce imperceptible attacks with a handful of patches. Thanks to our constraints, as the reviewer mentioned, we obtain more explainable and interpretable attacks.

---

> > ### Comment · Reviewer_H5Ak · 2021-11-29
> > **After rebuttal**
> >
> > Thanks for the authors' rebuttal and the improvements of the paper.
> > Most of my concerns were addressed in the response, except the overclaim on "physical" attacks without any demonstration. So considering the limitation and lack of theoretical contribution, I will keep my score.

---

> > > ### Author Response · Authors · 2021-11-30
> > > **Response to After rebuttal**
> > >
> > > Thank you for your feedback. We are willing to tone down further the claims about the relation of our attacks to physical attacks and mention them as future work. We would like to emphasize again that our work is the first to compute imperceptible multi-patch attacks using a handful of patches. Our approach is also the first to allow patches to overlap and thereby create complex patterns, while maintaining the simple formulation and a minimal number of degrees of freedom.

---

### Author Response · Authors · 2021-11-21
**Paper Revision**

We thank all the reviewers for their insightful reviews and helpful suggestions. We uploaded a modified version of our paper. We are happy to include more updates, if the reviewers recommend it. Here is a summary of the changes:

Section 2:

    - Fixed notations.

Section 3:

    - Improved the explanation of our approach and its novelty.

    - Shortened the gradient estimation and real-value relaxation paragraphs (to accommodate the changes within the page limit).

Section 4:

    - Provided better motivation for phrasing the problem as a SyGuS instance.

Section 5:

    - Emphasized the plots’ names using bold text.

Section 6:

    - Added all the related work suggested by the reviewers.

Appendix

    - Added experiments that show the effectiveness of our discrete optimization and adaptive optimization.

---

### Decision · Program_Chairs · 2022-01-20

**Decision:**

Reject

**Comment:**

This paper studies physical "adversarial programs" that allow an attacker to control a machine learning model by placing transparent patches on top of an image. The reviewers are split on this paper: while some reviewers like the work, others are concerned about the practicality, novelty, or utility of the attack.

Starting with novelty, reviewers raise valid concerns about how this approach is similar to prior attacks that generate programs. The authors respond here, but the overall question remains unanswered and it is not clear which of the new pieces this paper introduces are responsible for the success. (Would prior techniques have sufficed? If not what part of prior methods makes this not the case?)

For utility, the paper does not make a clear case of why it would be easier for an adversary to place N~=5 patches on top of an image as compared to other physical attacks (see especially Li et al. 2019 as a paper that deserves more than a sentence of comparison---why is this approach easier?).

One final comment raised by many reviewers is the fact that the title and setup to this paper heavily lean on the "physical" component of the evaluation, and yet the paper does not demonstrate anything physical. The authors rebuttal that the word "towards" absolves them of responsibility for trying an attack in the real world does not convince me; either the paper should attempt this attack in the physical world (and say if it works or if it doesn't) or make it clear from the top that the attack is going to be digital from the start, but motivated by the physical world. Prior accepted papers that include physical world in the title (e.g., Kurakin et al., Athalye et al., Li et al.) don't solve the problem completely, but at least run experiments in the physical world.